# The KEAP1-NRF2 System and Esophageal Cancer

**DOI:** 10.3390/cancers14194702

**Published:** 2022-09-27

**Authors:** Wataru Hirose, Hiroyuki Oshikiri, Keiko Taguchi, Masayuki Yamamoto

**Affiliations:** 1Department of Medical Biochemistry, Tohoku University Graduate School of Medicine, 2-1 Seiryo-Machi, Aoba, Sendai 980-8575, Japan; 2Department of Surgery, Tohoku University Graduate School of Medicine, Sendai 980-8575, Japan; 3Tohoku Medical Megabank Organization, Tohoku University, Sendai 980-8573, Japan; 4Advanced Research Center for Innovations in Next-Generation Medicine (INGEM), Tohoku University, Sendai 980-8573, Japan

**Keywords:** NRF2, KEAP1, esophagus, esophageal cancer, ESCC, cell competition

## Abstract

**Simple Summary:**

NRF2-activated or NRF2-addicted cancers show high incidence, especially in esophageal squamous cell carcinoma (ESCC). ESCC with high NRF2 expression is largely resistant to the current major treatments for ESCC and therefore shows a very poor prognosis. In order to develop effective treatments for NRF2-addicted esophageal cancers, the elucidation and understanding of the mechanistic basis of NRF2 function in NRF2-addicted cancer cells are critically important. This review summarizes the current knowledge of the KEAP1-NRF2 system and proposes three distinct approaches for the treatment of NRF2-addicted ESCC.

**Abstract:**

NRF2 (nuclear factor erythroid 2-related factor 2) is a transcription factor that regulates the expression of many cytoprotective genes. NRF2 activation is mainly regulated by KEAP1 (kelch-like ECH-associated protein 1) through ubiquitination and proteasome degradation. Esophageal cancer is classified histologically into two major types: esophageal squamous cell carcinoma (ESCC) and esophageal adenocarcinoma (EAC). ESCC harbors more genetic alterations in the KEAP-NRF2 system than EAC does, which results in NRF2 activation in these cancers. NRF2-addicted ESCC exhibits increased malignancy and acquisition of resistance to chemoradiotherapy. Therefore, it has been recognized that the development of drugs targeting the KEAP1-NRF2 system based on the molecular dissection of NRF2 function is important and urgent for the treatment of ESCC, along with efficient clinical screening for NRF2-addicted ESCC patients. Recently, the fate of NRF2-activated cells in esophageal tissues, which was under the influence of strong cell competition, and its relationship to the pathogenesis of ESCC, was clarified. In this review, we will summarize the current knowledge of the KEAP1-NRF2 system and the treatment of ESCC. We propose three main strategies for the treatment of NRF2-addicted cancer: (1) NRF2 inhibitors, (2) synthetic lethal drugs for NRF2-addicted cancers, and (3) NRF2 inducers of the host defense system.

## 1. Regulatory Mechanisms of NRF2 Activation

Nuclear factor erythroid 2-related factor 2 (NRF2 or NFE2L2) is a master transcription factor and regulates the expression of many cytoprotective genes [1]. The NRF2 protein level is regulated by proteasomal degradation through two E3 ubiquitin ligase complexes, kelch-like ECH-associated protein 1 (KEAP1)-Cullin 3 (CUL3) [2,3,4] and β-transducin repeat containing protein ligase (βTrCP)-Cullin 1 (CUL1) (Figure 1) [5,6,7]. In unstressed conditions, the NRF2 protein is maintained at a low level through these regulations utilizing protein degradation [3,8]. NRF2 activity can be induced in response to various stimuli, and the induction is predominantly exploited by the derepression mechanisms of protein degradation regulation processes [3]. Stresses originating from reactive oxygen species (ROS) and toxic chemicals (mostly electrophiles) are sensed by KEAP1. Under unstressed conditions, the KEAP1 homodimer binds NRF2 at two sites within the Neh2 domain, i.e., the DLG and ETGE motifs. Subsequently, the KEAP1-CUL3 ubiquitin E3 ligase complex ubiquitinates NRF2 quite efficiently. This process has been shown to occur in the cytoplasm [9], and ubiquitinated NRF2 is rapidly degraded through the 26S proteasome system. Upon exposure to oxidative or electrophilic stresses, specific reactive cysteine residues within KEAP1 are modified, leading to the attenuating of the ubiquitin ligase activity of the KEAP1-CUL3 complex and the ubiquitination of NRF2. Thus, NRF2 is stabilized, translocates into the nucleus and activates a set of cytoprotective genes [10]. The KEAP1-NRF2 system has been established as a major regulatory machine engaged in the response against oxidative and electrophilic stresses.

The βTrCP-CUL1 complex has also been identified to regulate cellular NRF2 levels [5,6]. This complex mainly operates within the nucleus. To be recognized by βTrCP, NRF2 needs to be phosphorylated by glycogen synthase kinase 3β (GSK3β) [5] at two specific serine residues in the Neh6 domain. Phosphorylated NRF2 is ubiquitinated and degraded through the proteasome system. A cascade of signal transduction has been clarified in which GSK3β is inhibited/phosphorylated by the activated phosphatidylinositol 3-kinase (PI3K)-AKT serine/threonine kinase (AKT) pathway, and PI3K/AKT activity is under the negative regulation of phosphatase and tensin homolog (PTEN).

The relative contributions of the KEAP1 pathway and βTrCP pathway have been elucidated. NRF2 does not accumulate in *Pten*-deleted mouse liver [11], indicating that βTrCP inactivation alone is insufficient to significantly activate NRF2. Analyses of NRF2 accumulation in *Pten*::*Keap1* double knockout mouse livers suggest that regulation by the KEAP1 pathway is predominant, while regulation by the βTrCP pathway is auxiliary and supportive in the regulation of NRF2. This model is nicely supported by an analysis of the newly established *Nrf2*^SA^ mutant mouse [12], which harbors replacements of two Ser residues at residues 335 and 338 by Ala. While the *Nrf2*^SA/SA^ mutation does not give rise to strong NRF2 activation in mouse tissues in vivo, simultaneous *Keap1* deletion markedly induces NRF2 activity. Severe growth retardation is provoked by the abnormal hyperkeratinization of the esophagus and forestomach.

In addition, the WD40 repeat-containing protein 23 (WDR23) is reported to directly bind the Neh2 domain of NRF2 and ubiquitinate NRF2 independently of KEAP1 by forming a complex with damage-specific DNA binding protein 1 (DDB1) and Cullin 4 (CUL4) [13]. This observation suggests that the WDR23-DDB1-CUL4 complex serves as the regulator of NRF2. Intriguingly, WDR23 was originally known to regulate basic-leucine zipper (b-ZIP) domain-containing protein skn-1 (skinhead-1 protein), which is the *C. elegans* ortholog of NRF2. The contribution of WDR23 to NRF2 regulation in human tissues needs to be clarified with further studies.

NRF2 forms a heterodimer with one of the three small musculoaponeurotic fibrosarcoma proteins (sMAF), i.e., MAFF, MAFG and MAFK [14,15]. The NRF2-sMAF heterodimer binds to the CNC-sMAF binding element (CsMBE), which is also known as the antioxidant responsive element (ARE) [16,17] or electrophile responsive element (EpRE) [18], in the regulatory regions of the target genes, such as glutamate-cysteine ligase catalytic subunit (*GCLC*), heme oxygenase-1 (*HO-1* or *HMOX1*), NAD(P)H quinone dehydrogenase 1 (*NQO1*), and thioredoxin reductase 1 (*TXNRD1*) [17]. In this way, upon exposure to oxidative or electrophilic stresses, the NRF2 protein is highly induced and exerts key regulatory influences over the cytoprotective function of the cells.

## 2. The Double-Edged Sword of NRF2 Activation in Cancer Cells

It is known that NRF2 protects the body from oxidative and electrophilic stresses. In this context, a breakthrough in the NRF2 field occurred in 2006. Two laboratories independently discovered somatic mutations in the *KEAP1* gene in lung cancer cells that disrupt the protein–protein interaction of KEAP1-NRF2, leading to the constitutive activation of NRF2 [19,20]. Soon after this discovery, somatic mutations in the *NRF2* gene that led to the constitutive activation of NRF2 were identified [21]. Since then, the constitutive activation of NRF2 has been found at a very high incidence in many cancers derived from various tissues, driven not only by somatic mutations but also by various other mechanisms (see below) [22,23,24]. We refer to these cancer cells with high NRF2 activity levels as NRF2-activated cancers or NRF2-addicted cancers [25]. Owing to the cytoprotective and metabolic reprogramming functions of NRF2, NRF2-addicted cancers acquire a marked advantage in proliferation and resilience against chemoradiotherapy, resulting in a poor prognosis [21,26]. Therefore, NRF2 activation is often referred to as a ‘double-edged sword’ or by similar phrases [27,28,29]. As described in Section 9.3, the strategy targeting NRF2 activation has been proposed for the NRF2-addicted cancer treatments [30].

From a mechanistic viewpoint, NRF2 activation in cancer cells is mainly elicited by the dysfunction of protein–protein interactions in the NRF2-KEAP1-CUL3 ubiquitin E3 ligase complex. Somatic mutations in the *NRF2*, *KEAP1* or *CUL3* genes have been well characterized [31]. Intriguingly, mutations in *NRF2* accumulate in two KEAP1-interacting motifs, i.e., the DLG and ETGE motifs within the N-terminal Neh2 domain [21,25,32,33]. The extended DLG (DLGex) motif spans amino acid positions 17 to 51, while the ETGE motif is located at positions 77 to 82 [34]. As described above, these somatic mutations are gain-of-function mutations of NRF2.

In addition to somatic mutations, several other causes of NRF2 activation have been identified. Exon skipping of *NRF2* exon 2 and/or exon 3 has been reported in lung cancer and head and neck cancers [35]. The removal of *NRF2* exon 2 abolishes its ability to bind to KEAP1 and activates NRF2. Epigenetic modification, i.e., hypermethylation in the promoter region of the *KEAP1* gene, has also been identified to constitutively activate NRF2 in lung and renal cancers [36,37]. The modification of KEAP1 cysteine residues by oncometabolites such as fumarate and itaconate has also been reported to activate NRF2 [38,39,40]. All of these factors lead to constitutive NRF2 activation and contribute to growth, malignant transformation and chemoradiotherapy resistance.

An intriguing and unexpected mechanism is the disruption of the KEAP1-NRF2 interaction through the accumulation of the autophagy chaperone p62 [41]. p62 is encoded by the sequestosome 1 (*SQSTM1*) gene [42] and retains a phylogenetically conserved STGE motif within the KEAP1-interacting domain [41]. After the phosphorylation of serine residues in the motif, the pSTGE motif mimics the ETGE motif in NRF2, and its affinity for KEAP1 is comparable to that of ETGE and higher affinity than that of the DLGex motif [43]. p62 accumulation causes NRF2 activation by inhibiting the KEAP1-NRF2 interaction via a Hinge-and-Latch mechanism [10]. This mechanism has been verified using a mouse model with knockout of *Atg7*, an essential gene for autophagy [44]. Notably, it has been found that hepatocellular adenoma is provoked by p62-mediated NRF2 activation in liver-specific *Atg7* knockout mice [45]. Consistent with the findings in the mouse model, in human hepatocellular carcinoma, p62- and KEAP1-positive aggregates are abundant, suggesting that successive NRF2 activation contributes to the development of hepatocellular carcinoma [45]. It has also been reported that in human esophageal squamous cell carcinoma (ESCC) patients, NRF2 activation and phospho-p62 accumulation are associated with radioresistance [46].

## 3. Functions of NRF2 in the Esophagus and Esophageal Cancers

NRF2 appears to be related to the differentiation [47], development [48], barrier function [49] and metabolism [50] of the esophagus. The comparison of *NRF2* expression in various tissues revealed that the esophagus expresses *NRF2* at the highest level (Human Protein Atlas; https://www.proteinatlas.org, accessed on 6 May 2022) [51]. In detail, *NRF2* expression in the esophagus was 5.4-fold higher than that in the lowest-expressing tissue (pancreas) and 1.5-fold higher than that in the second-highest-expressing tissues (stomach and liver) (Figure 2a). In contrast, the *KEAP1* level is extremely high in skeletal muscle compared with other human tissues (Figure 2b).

The *K**EAP1* level in the esophagus is comparable to that in the tongue and liver. While NRF2 activation is meticulously regulated at the protein level, the high *NRF2* mRNA level in the esophagus suggests that the esophageal epithelium has the ability to express high levels of NRF2 at the protein level. In fact, severe hyperkeratosis in the esophageal epithelium is observed as the most characteristic phenotype in *Keap1* systemic knockout (*Keap1*^–/–^) mice [52].

Apart from its function in the normal esophageal epithelium, NRF2’s contribution to the development and progression of esophageal cancer has been a topic of particular interest in recent years. Esophageal cancer is one of the most aggressive tumors in humans and is the sixth leading cause of cancer-related death [53]. Esophageal cancer is classified histologically into esophageal adenocarcinoma (EAC) and ESCC, which show distinct epidemiological and genomic characteristics [54] (Figure 3).

EAC is predominant in North America and Europe [55]. Obesity increases the risk of Barrett’s esophagus and EAC due to the increase in intragastric pressure or hiatal hernia [56]. In Barrett’s esophagus caused by gastroesophageal reflux disease (GERD), normal squamous mucosa in the lower thoracic to abdominal esophagus is replaced with metaplastic columnar epithelium, a precancerous condition preceding EAC (Figure 3). In EAC, *TP53* mutations are found at a high level, approximately 70% [57]. Genomic alterations in the KEAP1-NRF2 system are rarely found in EAC [57], but mutation/silencing of *CDKN2A* and amplification of *ERBB2* and *VEGFA* are found frequently. EAC shows genomic characteristics similar to the most common chromosomal instability type of gastric adenocarcinoma [58].

In contrast, two studies reported that NRF2 is constitutively activated in EAC in the absence of somatic mutations in *NRF2* or *KEAP1* [59,60]. This finding is interesting if we consider the finding that NRF2 is activated in the esophageal epithelium with GERD, which suggests that NRF2 plays a cytoprotective function in GERD [49]. Thus, the contribution of NRF2 activation to EAC development is currently unclear. Further examinations utilizing sophisticated in vivo models of EAC are necessary for this purpose.

## 4. Frequently Activated NRF2 in ESCC

The incidence of ESCC is more prevalent in Asia, Africa, and South America than in Europe and North America [53]. Most cases of ESCC occur in the middle thoracic esophagus region, and ESCC is found to be highly associated with smoking and alcohol consumption [61] (Figure 3). The genomic characteristics of ESCC resemble those of squamous cell carcinoma of the lung and head-and-neck [54]. *TP53* mutations are commonly found in esophageal cancers. In particular, *TP53* mutations are found in more than 80% of ESCC cases, while *TP53* mutations are found in approximately 70% of EAC cases.

*NRF2* is one of the frequently mutated genes in ESCC, and 10–22% of ESCC cases retain somatic mutations in the *NRF2* gene [21,62]. In addition to *NRF2*, somatic mutations are also found in *MLL2*, *ZNF750*, *NOTCH1* and *TGFBR2* [63,64,65,66,67,68]. Genomic amplifications of *CCND1* and *SOX2*/*TP63* and deletions of *CDKN2A* have also been reported in ESCC cases. For NRF2 activation, a variety of genetic alterations in the members of the KEAP1-NRF2 system or related molecules should also be considered. These abnormalities include somatic mutations of *KEAP1*, somatic mutations or deletions of *CUL3,* and deletions of *ATG7*, which occur in 2–4%, 1–5% and 6% of ESCC cases, respectively [54]. Furthermore, the activation of the PI3K-AKT pathway is frequently found in ESCC. For instance, *PTEN*, one of the members of this pathway, harbors somatic mutations and deletions in 2–5% and 3% of cases, respectively. Another member, *PIK3CA,* exhibits somatic mutations, amplifications and overexpression in 5–10%, 10–41%, and 50% of cases, respectively [63,64,66,69,70]. Activating mutations or abnormalities in the PI3K-AKT pathway in ESCC are particularly intriguing, as these contribute to the inactivation of the downstream GSK3β, and GSK3β is known to induce NRF2 activation [5].

Of the many intriguing observations that can be extracted from the TCGA Pan-Cancer Atlas using cBioPortal [71,72] (http://cancergenome.nih.gov/, accessed on 6 May 2022), one of the most surprising findings is that ESCC is the cancer with the highest frequency of genomic alterations, which include somatic mutations, amplifications and deletions (Figure 4). *NRF2* somatic mutations were originally found in squamous cell carcinoma (SCC) in the lung [19], and it had previously been considered that lung SCC must be the cancer type with the highest prevalence of *NRF2* somatic mutations.

Based on comprehensive analyses of DNA methylation, mRNA expression, copy number variation and microRNA expression data by TCGA Research Network [54], ESCC is clustered into 3 subtypes: ESCC1, ESCC2 and ESCC3. ESCC1 is the major subtype (56%) and frequently harbors genomic alterations in the KEAP1-NRF2 system and related members. For instance, 30% of ESCC1 cases harbor genomic alterations in *NRF2*, 6% harbor alterations in *KEAP1*, and 10% harbor alterations of *CUL3*. ESCC2 and EAC retain, albeit at a low frequency, genomic alterations in the KEAP1-NRF2 system.

The frequency of NRF2-activated cancers may be much higher than that of cancers bearing gene alterations in the KEAP1-NRF2 system and related members. Additional mechanisms may lead to NRF2 activation in ESCC. Since NRF2 activation confers chemoradiotherapy resistance and poor prognosis to ESCC cases [21,26,73,74,75], it is important to clarify the contribution of NRF2 to cancer biology and medicine.

## 5. NRF2-Addicted ESCC Cell Lines

Two series of ESCC cell lines have been established in Japan. One is the TE series of ESCC cell lines. The first paper describing the TE cell line was published in 1979 by Nishihira et al. at Tohoku University [76]. The other ESCC cell line is the KYSE series, which was reported in 1992 by Shimada et al. at Kyoto University [77]. Other ESCC cell lines have also been established and used to study various aspects of ESCC. Somatic mutations eliciting NRF2 activation and their consequences have been studied utilizing these cell lines.

We evaluated ESCC cell lines for NRF2 expression levels and the presence of somatic mutations in related genes. The results are summarized in Table 1. ESCC cell lines can be classified into two groups based on the levels of NRF2 activation status, i.e., a high-level NRF2-expressing group and a normal-level NRF2-expressing group [21,73,78,79,80,81]. This classification can be seen in clustering analysis using RNA-sequencing (RNA-seq) data [82] as well as somatic mutations in the COSMIC database (Catalogue of Somatic Mutations in Cancers; https://cancer.sanger.ac.uk/cell_lines) [83]. In this search, the TE and KYSE series of cell lines were found to contain almost comparable numbers of NRF2 high-level and normal-level cell lines (Table 1). Because of the clinical observations summarized above, we expected that NRF2 high-level cell lines would be derived more from poorly differentiated ESCC than from highly differentiated ESCC. However, the NRF2 activity status appears not to be closely related to the cancer cell differentiation stage of the original ESCC (Table 1).

## 6. Esophageal Phenotype in Mouse Models with Genetic Modification of *Nrf2* or *Keap1*

Studies on the physiological and pathological contributions of the KEAP1-NRF2 system in the esophagus have been developed using genetically modified mice (Table 2). Two lines of *Nrf2* systemic knockout (*Nrf2*^–/–^) mice were generated in the 1990s [85,86] and have been utilized in various analyses. *Nrf2*^–/–^ mice are sensitive to chemical carcinogenesis, as phase II detoxication enzymes are under NRF2 regulation, so their expression is downregulated and not induced in response to carcinogens [86]. In fact, *Nrf2*^–/–^ mice develop many more tumors in the upper aerodigestive tract than wild-type mice upon the administration of a chemical carcinogen, 4-nitroquinoline-1-oxide (4NQO) [87].

We also generated *Keap1* systemic knockout (*Keap1*^–/–^) mice that serve as an NRF2-activated genetic model by inserting the *LacZ* gene into *Keap1* exon 2 (Figure 5a) [52]. Unexpectedly, *Keap1*^–/–^ mice develop hyperkeratosis of the esophageal and forestomach epithelium and die before weaning because of the lack of food intake. The lethality and hyperkeratosis is nicely rescued by simultaneous systemic *Nrf2* knockout (*Keap1*^–/–^::*Nrf2*^–/–^) [52] or keratinocyte-specific *Nrf2* knockout (*Keap1*^–/–^::*Nrf2*^flox/flox^::*Keratin5-Cre*) [90], clearly demonstrating that esophageal hyperkeratosis is provoked by constitutive and massive NRF2 activation as a consequence of *Keap1* deletion. *Keap1*^–/–^ mice are rescued from juvenile lethality, and hyperkeratosis is largely avoided in *Nrf2* heterozygous compound mutant mice (*Keap1*^–/–^::*Nrf2*^+/–^) [89], indicating that massive *Nrf2* expression is required to provoke severe hyperkeratosis of the esophagus (Table 2).

We found that while the wild-type *Keap1* transgene (*Tg-Keap1*^WT^) is able to rescue esophageal hyperkeratosis in *Keap1*^–/–^ mice [91], *Keap1* point mutant transgenes (*Tg*-Keap1 mutant), including C273&288A, G364C and G430C, cannot rescue hyperkeratosis and lethality. These results indicate that these amino acid residues are critical for KEAP1 function to repress NRF2 [91,92]. In fact, cysteine 273 (C273) of KEAP1 serves as a sensor for various electrophiles, and C288 serves as a sensor for 15d-prostaglandin J_2_, 4-hydroxynonenal and arsenite [96]. G364C and G430C mutations were originally identified in human lung cancer patients [19] but have not been found in ESCC to date. Intriguingly, *Keap1*^–/–^::*Tg-Keap1*^WT^::*Tg-Keap1*^mutant^ mice show hyperkeratosis and juvenile lethality (Table 2), whereas *Keap1* heterozygous knockout mice (*Keap1*^+/–^) have no obvious changes in the esophagus [52,93]. These data suggest an intriguing mechanism. Since KEAP1 forms a homodimer and binds one molecule of NRF2, mutant KEAP1 may form a heterodimer with wild-type KEAP1, which lacks ubiquitin ligase activity. Therefore, KEAP1 mutants may have a dominant-negative effect in ESCC cell lines harboring heterozygous *KEAP1* mutations (Table 1).

To further analyze the consequences of *Keap1* deletion more specifically, two lines of conditional knockout mice were generated: the *Keap1*^floxA^ mouse line [97] and the *Keap1*^floxB^ mouse line [98] (Figure 5a). The *Keap1*^floxA^ allele has two loxP sites on both sides of *Keap1* exons 4–6 [97] and green fluorescent protein (GFP) downstream of exon 6 (Figure 5a). From this allele, we found that *Keap1* mRNA expression is decreased to 20% of that from the wild-type allele, even in the absence of Cre recombination [93,98] (Figure 5b). As *Keap1*^floxA/–^ mice show milder hyperkeratosis than *Keap1*^–/–^ mice, *Keap1*^floxA/–^ mice can survive with the remaining *Keap1* mRNA expression, which is approximately 10% of the level of wild-type mice [93]. We have been capitalizing this genetically hypomorphic expression of KEAP1 and utilizing the *Keap1*^floxA/floxA^ or *Keap1*^floxA/–^ lines of mice as *Keap1* knockdown models, which are viable models showing the systematic expression of high-level NRF2.

Consistent with the phenotype in *Keap1*^–/–^ mice, the hyperkeratosis of the esophagus and forestomach in *Keap1*^floxA/–^ mice is also diminished by the transgenic expression of wild-type KEAP1 (*Tg-Keap1*^WT^) [93]. Meanwhile, *Keap1* mRNA expression in *Keap1*^floxA/floxA^ mice completely disappeared in the presence of K5-Cre recombinase (Figure 5b), so keratinocyte-specific *Keap1* knockout (*Keap1*^floxA/floxA^::*K5Cre*) mice died before weaning due to severe hyperkeratosis of the esophagus, similar to *Keap1*^–/–^ mice [93]. As expected, *Keap1*^+/floxA^::*K5Cre* mice showed no obvious change in the esophagus, similar to *Keap1*^+/–^ mice. Upon exposure to 4NQO, a chemical carcinogen, *Keap1*^floxA/–^ mice develop many fewer tumors than wild-type mice [87]. We surmise that this is likely due to the high level of expression of cytoprotective genes, but it is possible that mechanical protection by hyperkeratosis or thickened epithelium in the esophagus also plays a role.

Biswal and colleagues generated the *Keap1*^floxB^ line of mice in which two loxP sites flank exons 3–4 of *Keap1* (Figure 5a). Unlike the *Keap1*^floxA^ allele, the *Keap1*^floxB^ allele retains intact *Keap1* mRNA expression levels in the absence of Cre recombinase [94,98]. After Cre recombination, *Keap1* expression is almost completely knocked out. These two *Keap1*^flox^ mouse lines exhibit intriguing phenotypic differences based on graded *Keap1* expression. Thus, we are able to impose subtle changes in *Keap1* gene expression depending on the purpose.

## 7. Cell Competition in Esophageal Epithelium and the Fate of NRF2-Deleted Cells

The existence of stem cells in the esophageal epithelia is still controversial [99]. While one group insists that the esophageal epithelium is maintained by uniformly distributed progenitor cells in the basal layer [100], other groups propose that specific esophageal epithelial stem cells exist in the basal layer, which is composed of one to four cell layers in humans and only one layer in mice. In the former theory, each progenitor cell attaches to the basement membrane and shows equal self-reproducibility and the ability to contribute to the maintenance of esophageal epithelium homeostasis by supplying differentiated epithelial cells. The latter theory assumes the existence of esophageal stem cells with high-level expression of integrin α6 (ITGA6), ITGB4 and CD73 [101], KRT15 [102] or COL17A1 and KRT15 [103].

Related to this issue, the presence of a cell competition mechanism within the esophageal epithelium has been revealed by means of lineage tracing experiments using fluorescent reporter mice and mathematical models [100,104,105,106,107,108,109,110]. This mechanism is particularly interesting if we consider the fact that various somatic mutations accumulate and cell clusters harboring those mutations emerge in a patchy or island-like manner in the homogenous population of the esophageal squamous epithelium. In this model, ‘winner’ clones that acquire a competitive advantage to the surrounding ‘loser’ clones expand their cell population. In fact, it has been shown that epidermal cells with high expression of COL17A1 become ‘winners’ [111]. An intriguing observation here is that the expression of hemidesmosome constituents, including COL17A1, ITGA6 and ITGB4, is closely related to both esophageal stemness [101,103] and cell competition [105].

Given the high incidence of somatic mutations in the *NRF2* gene in ESCC, we hypothesized that the loss of NRF2 activity might influence cell competition in the esophageal epithelium. To address this hypothesis, we designed a CreERT2-based conditional deletion of the *Nrf2* gene. The CreERT2 system has been used in numerous lineage tracing studies to induce the tissue-specific and tamoxifen-inducible expression of Cre recombinase [112,113]. The recombination efficiency of this system is limited within 30–80%, which also varies among Cre-expressing vector constructs, animals, genotypes, tissues, cell types and tamoxifen doses [114,115]. Owing to these characteristics, we successfully established an in vivo model in which NRF2-deleted cells coexisted with NRF2-normal cells [88]. Features of these coexistent cells were monitored after NRF2 deletion in the mouse esophageal epithelium by crossing *Keratin5-CreERT2* (*K5CreERT2*) mice and administering tamoxifen.

In *Nrf2*^flox/flox^::*K5CreERT2* mice, approximately 50% of *Nrf2* DNA in the esophageal epithelium is recombined, and NRF2-deleted cells emerge segmentally [88]. While NRF2-deleted cells resided and were maintained in the epithelium under normal conditions, we found that these cells were selectively eliminated from the epithelium upon exposure to the chemical carcinogen 4NQO (Figure 6a). That is, NRF2-normal cells survive, while NRF2-deleted cells disappear in the presence of the chemical carcinogen 4NQO. One plausible interpretation of this result is that NRF2-deleted cells are lost in competition with NRF2-normal cells in the esophageal epithelium in the presence of the chemical carcinogen 4NQO.

## 8. Fate of NRF2-Activated Cells in the Esophageal Epithelium

We also hypothesized that a gain of NRF2 activity may influence cell competition in the esophageal epithelium. To address this hypothesis, we designed a CreERT2-based conditional deletion of the *Keap1* gene and successfully established an in vivo model in which KEAP1-deleted (i.e., NRF2-activated) epithelial cells coexist with KEAP1-normal (i.e., NRF2-normal) cells in the esophageal epithelium [95]. The features of these coexistent cells were monitored after KEAP1 deletion/NRF2 activation in the mouse esophageal epithelium after the administration of tamoxifen. To our surprise, in the esophagus of *Keap1*^floxB/floxB^::*K5CreERT2* mice, KEAP1-deleted (i.e., NRF2-activated) cells were selectively and rapidly eliminated from the esophageal epithelium even under normal conditions (Figure 6b). KEAP1-deleted/NRF2-activated cells exhibit decreased COL17A1 expression and commit to cell differentiation. Thus, the cells lose their competitive advantage against KEAP1-normal/NRF2-normal cells in the epithelium. An intriguing hypothesis that emerges from these observations is that KEAP1-deleted/NRF2-activated cells may be routinely generated in the epithelium due to somatic mutations caused by the toxic substances passing through the esophagus but then be eliminated regularly through cell competition in the esophageal epithelium.

An important question here is the mechanism by which NRF2-activated or NRF2-addicted cancers are generated in light of the cell active competition model. We surmise that multiple pathways or events are involved in the formation of NRF2-addicted esophageal cancers in addition to NRF2 activation. The study showing that *NRF2* mutation is a late event during ESCC evolution supports this hypothesis [116]. As described in Section 3 and Section 4, *TP53* mutations are the most common mutations in ESCC. In fact, mice cells with the *Tp53*^R245W^ mutation (equivalent to the *TP53*^R248W^ mutation in humans) are shown to become winners and expand clonally in the competition between the esophageal epithelium and epidermal cells [107,108]. Esophageal carcinoma retains a variety of driver candidates in addition to TP53. Therefore, somatic mutations in these genes, including *TP53*, may be the tumor initiation mutations required for NRF2-activated cells to become winner cells and ultimately develop into cancer cells. We hypothesize that NRF2 activation may play a role in favoring the survival/expansion of winner *TP53-* or other mutation-bearing cells.

## 9. NRF2 as a Therapeutic Target for Cancers

It has been shown that NRF2-addicted cancers in various tissues or organs, including NRF2-addicted ESCC, are resistant to current mainstream cancer therapies. Some studies have reported that NRF2-addicted ESCC is resistant to chemoradiotherapy using cisplatin and fluorouracil, the major treatments for ESCC [26,46]. Even immune checkpoint inhibitors, including nivolumab and pembrolizumab, are not effective, especially in NRF2-addicted non-small cell lung cancer (NSCLC). NRF2-addicted NSCLC shows a poorer prognosis than NRF2-normal cancer [117]. To overcome NRF2-addicted cancers that are resistant to general or mainstream treatments, new therapies targeting NRF2 appear to be necessary. To this end, we propose three strategies that target either NRF2-addicted cancer cells or host defenses for the effective treatment of NRF2-addicted cancers (Figure 7).

### 9.1. NRF2 Inhibitors to Treat NRF2-Addicted Cancers

It seems reasonable to consider the development of NRF2 inhibitors as an approach against NRF2-addicted cancers. Through chemical library screening utilizing an ARE-Luc reporter, we identified halofuginone, a derivative of the plant alkaloid febrifugine, as a specific inhibitor of NRF2 [80]. Mechanistic studies have revealed that halofuginone induces the amino acid starvation response by inhibiting prolyl-tRNA synthetase activity [118], resulting in the depletion of the NRF2 protein, as NRF2 shows very rapid turnover with a half-life of less than 18 min [3,8]. Halofuginone has been widely used as an antibiotic in animals [119]. In fact, halofuginone enhanced the anticancer effects of cisplatin in both in vitro and in vivo experiments using KYSE70, a cell line derived from NRF2-addicted ESCC [80]. Halofuginone has already been used in phase II human trials as a therapy for AIDS-related Kaposi sarcoma [120]. While halofuginone exhibits some adverse effects, including a reduction in red blood cells and white blood cells, we have developed a nanomedicine technology in which halofuginone is encapsulated into micelles [121]. Halofuginone-micelles show much reduced side effects with regard to suppressing hematopoietic and immune cells [121], demonstrating that halofuginone-micelles are promising NRF2 inhibitors for the treatment of NRF2-addicted ESCC.

Along this line, Biswal and colleagues identified ML385 as an NRF2 inhibitor through high-throughput screening [122]. ML385 is a small molecule that directly inhibits NRF2 DNA binding through the Neh1 domain and blocks NRF2 transcriptional activity. ML385 shows a specific antitumor effect for NRF2-addicted NSCLC with *KEAP1* mutation both in vitro and in vivo [122,123], indicating that ML385 is another important drug candidate for the treatment of NRF2-addicted ESCC. In addition, several other phytochemicals have been reported as NRF2 inhibitors [124,125,126]. Further development of these drugs is expected.

### 9.2. Synthetic Lethal Drugs to Treat NRF2-Addicted Cancers

The development of synthetic lethal anticancer drugs is an alternative approach for the treatment of NRF2-addicted cancers. This approach exploits the ability of NRF2 to induce a set of drug metabolizing enzymes [127,128]. Therefore, in this scenario, a set of drug-metabolizing enzymes that are highly activated in NRF2-addicted cancers transform anticancer prodrugs into active anticancer drugs. This approach and the identified synthetic lethal drugs harbor several critical advantages. For instance, nonmutant cells within the patient will be insensitive to the treatment, so the drugs should have a large therapeutic window with limited adverse effects.

Through synthetic lethal drug screening of an off-patent drug library utilizing isogenic cell lines, mitomycin C was identified as a promising candidate [81]. Mitomycin C is a DNA alkylating agent and shows enhanced toxicity specific to NRF2-addicted cancer cells. Mitomycin C contains a quinone structure that is bioactivated through cytochrome P450 reductase (P450R or CYPOR) and NQO1, which are NRF2-target gene products [81]. Therefore, mitomycin C is specifically activated by aberrant NRF2 activation. As an already approved clinical drug for cancers in the head and neck, stomach, colon and anus, pancreas, lung, cervix, bladder, and breast [129], mitomycin C is expected to be the first practical and economical drug for NRF2-addicted ESCC.

The geldanamycin-derived HSP90 inhibitors 17-AAG, 17-DMAG and IPI-505 have also been identified as synthetic lethal drugs for NRF-addicted cancer cells [127]. For this group of drugs, NQO1 acts to metabolize quinone compounds to hydroquinone and convert less-toxic prodrugs into actively toxic anticancer drugs specifically in NRF2-activated cells. These drugs have been used in many clinical trials but were withdrawn primarily due to toxicity issues [130,131,132]. While position 19-substituted geldanamycins have been developed to ameliorate this toxicity by blocking the reaction of the compounds with biological nucleophiles [133], we would like to propose an alternative approach in which geldanamycins are used for the treatment of NRF2-addicted cancers based on the synthetic lethal mechanism.

### 9.3. NRF2 Inducers to Target the Host Defense System

As a completely distinct strategy for treating NRF2-addicted cancers, we propose NRF2 inducers targeting the host microenvironmental defense. In a series of analyses utilizing chemical carcinogenesis of the mouse lung and esophagus, we identified that systemic activation of NRF2 in mice, both genetic and pharmacological, improves the survival of the mice with NRF2-addicted cancers [87,98,134]. These analyses converge to the conclusion that NRF2 activation in microenvironment cells suppresses tumor progression and is beneficial for the treatment of NRF2-addicted cancers. The concept of this strategy may be symbolized as ‘Fighting Fire with Fire’. The activation of NRF2 in microenvironmental cells may contribute to cells not only overcoming the adverse effects of anticancer drugs but also protecting the anticancer immunity of the host from the immune suppressing effect elicited by NRF2-activated cells. In this regard, it should be noted that NRF2 inducers are used in humans. For instance, sulforaphane [135,136,137] is used as a cytoprotective dietary supplement, and dimethyl fumarate [138] is used for the treatment of relapsing multiple sclerosis. Bardoxolone methyl [139,140] is now in clinical trials for diabetic kidney disease and other diseases.

### 9.4. Others

Several studies have focused on metabolic activities in NRF2-addicted cancers. The dual inhibition of glycolysis and glutaminolysis by the mTOR inhibitor sapanisertib (also named MLN0128, INK128 or TAK-228) and the glutaminase inhibitor telaglenastat (CB-839) to target NRF2-addicted NSCLC is currently in a clinical trial [141,142]. The suppression of glycolysis by mTOR inhibitors elicits metabolic adaptation thorough glutaminolysis in cancer cells that consume abundant glucose [143]. As NRF2-addicted cancers exhibit enhanced glutamine dependence, the inhibition of glutaminolysis is expected to suppress the proliferation of NRF2-addicted cancers. Therefore, the combination of sapanisertib and telaglenastat may become a therapy in NRF2-addicted cancers.

The development of microRNAs (miRNAs) may become a potential therapy for cancers. It has been shown that NRF2 in ESCC is downregulated by miR-507, -634, -450a, and -129-5p, or activated through the downregulation of KEAP1 by miR-432-3p [144,145]. Therefore, these miRNAs targeting NRF2 may emerge as a potential therapy to treat NRF2-addicted ESCC.

## 10. Toward Clinical Use of NRF2 Target Drugs

As described above, genetic alterations in the KEAP1-NRF2 system occur in more than 30% of ESCC cases. One of the important issues remaining to be addressed to achieve progress in NRF2-addicted cancer-specific treatments is how to efficiently and precisely identify the NRF2-activated status in clinical sites. NRF2 activation is generally evaluated according to the nuclear accumulation of NRF2 proteins and the upregulation of target genes. Therefore, multiple approaches to detect NRF2 activation have been used for human ESCC samples at the experimental stage.

One classic approach is the immunohistochemical analysis of tumor tissue specimens for the nuclear NRF2 protein [26,46,74,75,78,146] In the immunohistochemical analysis, higher NRF2 expression in ESCC than in the matched normal tissue in the same case is generally considered to signify the detection of an NRF2-addicted cancer. Utilizing this approach, it has been identified that NRF2 expression in cancers is associated with tumor stage and clinical outcome [73]. Similarly, accumulating lines of evidence support the relationship between nuclear NRF2 expression and survival [26,74,147,148].

An alternative approach is to use cancer genome analysis and challenge the detection of *NRF2* gene alterations in cancers [21,33,62,75]. In fact, somatic mutations in the *NRF2* gene are shown to be associated with NRF2 accumulation in the nucleus [75] and poor prognosis in ESCC patients [21]. However, there are causes of NRF2 activation other than *NRF2* genomic alterations, and these causes need to be examined. Such alternate causes include exon skipping, epigenetic modification and oncometabolites (*see* Section 2), which should not be overlooked in the analysis of NRF2-addicted cancers.

Comprehensive examination of NRF2 target gene mRNA expression by quantitative PCR or RNA sequencing analysis is an attractive approach toward identifying NRF2-addicted ESCC [73,78]. However, there remain clinical hurdles for the routine examination of RNA expression in tumor specimens. In this regard, the establishment of clinical phenome centers as well as further technological advances are anticipated. Most of all, as a less invasive approach, liquid biopsy using a small volume of blood or urine samples for the detection of specific molecules derived from NRF2-addicted cancers is highly anticipated, although the development of these technologies has been quite challenging to date.

## 11. Conclusions

It has been emerging that NRF2 activation is one of the major phenotypes that is intimately related to poor prognosis in ESCC. Three main approaches have emerged in the treatment of the NRF2-addicted cancers. Further clinical research and/or trials with these approaches will tell us how we can overcome these NRF2-activated cancers. We fully expect that new treatments for NRF2-addicted ESCC will improve the prognosis of the ESCC patients in the near future.

## Figures and Tables

**Figure 1 cancers-14-04702-f001:**
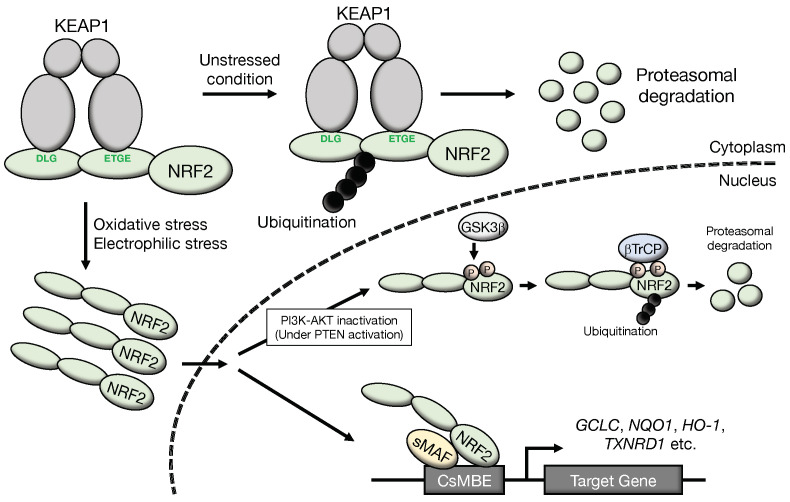
Molecular mechanisms of NRF2 activation: NRF2 degradation is accelerated through ubiquitination of NRF2 by the KEAP1-CUL3 complex and βTrCP-CUL1 complex. Under unstressed conditions, the KEAP1 homodimer binds NRF2 at two sites within the Neh2 domain, i.e., the DLG and ETGE motifs, and NRF2 is ubiquitinated efficiently. Because NRF2 is rapidly degraded by the proteasome system, the NRF2 protein is maintained at a very low level in unstressed conditions. Upon exposure to oxidative or electrophilic stresses, specific reactive cysteine residues within KEAP1 are modified. This modification attenuates the ubiquitin ligase activity of KEAP1. Thus, NRF2 is stabilized, translocates into the nucleus, and activates a set of cytoprotective genes. On the other hand, the βTrCP-CUL1 complex also ubiquitinates NRF2. To be recognized by βTrCP, NRF2 is phosphorylated by GSK3β at two specific serine residues in the Neh6 domain. Phosphorylated NRF2 is ubiquitinated by the βTrCP-CUL1 ubiquitin E3 ligase complex and degraded through the proteasome system. A cascade of signal transduction has been clarified in which GSK3β is inhibited/phosphorylated by the activated PI3K-AKT pathway, which is under the negative regulation of PTEN. CsMBE, CNC-sMAF binding element.

**Figure 2 cancers-14-04702-f002:**
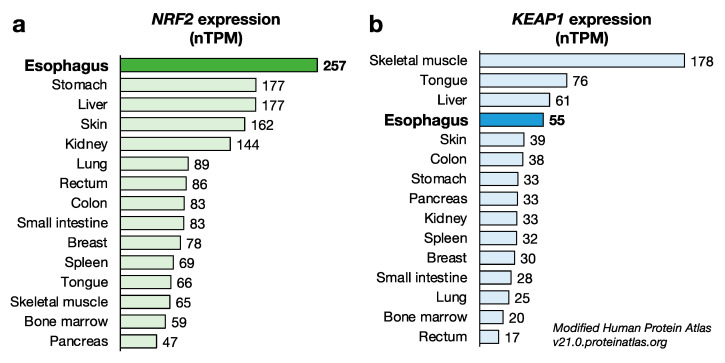
Expression levels of *NRF2* and *KEAP1* protein-coding transcripts in various tissues. (**a**) Expression level of *NRF2* protein-coding transcripts in various tissues. Of note, the *NRF2* expression level in the esophagus was the highest among the tissues examined. (**b**) Expression level of *KEAP1* protein-coding transcripts in various tissues. The *KEAP1* expression level in the esophagus is comparable to that in the tongue and the liver. Data were obtained from Human Protein Atlas v21.0.proteinatlas.org. nTPM: normalized protein-coding transcripts per million.

**Figure 3 cancers-14-04702-f003:**
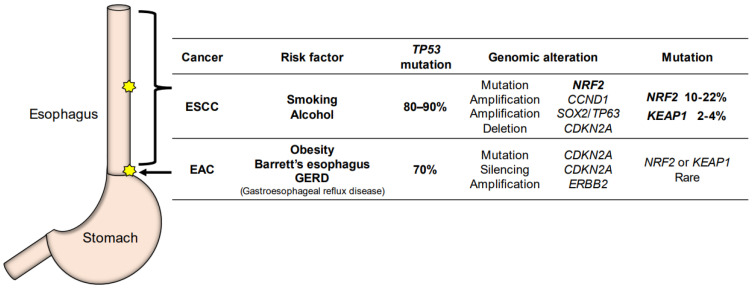
The characteristic difference between ESCC (esophageal squamous cell carcinoma) and EAC (esophageal adenocarcinoma). Most cases of ESCC occur in the middle thoracic esophagus, and ESCC is highly associated with smoking and alcohol consumption. Somatic mutations of *NRF2* are often found in ESCC. EAC occurs in the lower thoracic esophagus, down to the esophagogastric junction. EAC originates from Barrett’s esophagus and GERD (gastroesophageal reflux disease). Genomic alterations in the KEAP1-NRF2 system are rarely found in EAC.

**Figure 4 cancers-14-04702-f004:**
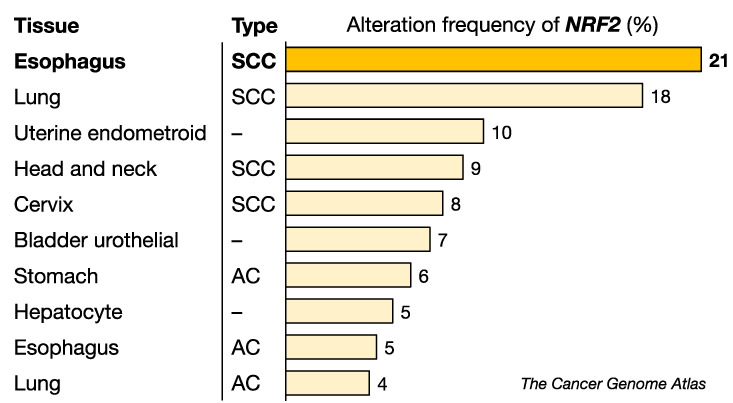
Alteration frequency of the *NRF2* gene in various cancers. ESCC has the highest alteration frequency among the cancers examined. *NRF2* mutations appear to be more common in SCC than in AC. SCC, squamous cell carcinoma, AC, adenocarcinoma. Data were obtained from the TCGA dataset.

**Figure 5 cancers-14-04702-f005:**
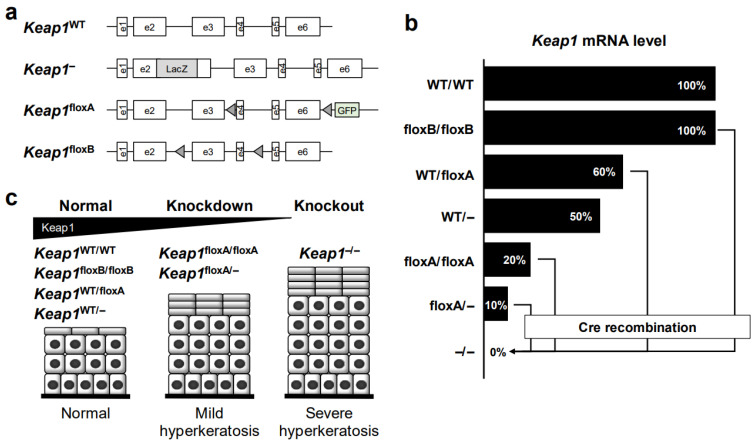
Genetically modified *Keap1* and esophageal phenotypes in mice. (**a**) Construction of *Keap1*-modified genes in mice. Diagrams of wild-type (WT), systemic knockout (–/–), and *Keap1*^flox^ mice (two lines) are shown. (**b**) The expression level of *Keap1* mRNAs in *Keap1* genetically modified mice. The *Keap1* mRNA expression from the *Keap1*^floxB^ allele is comparable with that of the WT allele in the absence of Cre recombination. The expression in the homozygous *Keap1*^floxA^ allele is decreased to 20% of that of the WT allele, even in the absence of Cre recombination. *Keap1* in floxA/floxA, floxB/floxB or floxA/– appears to be substantially deleted in the presence of Cre recombination. (**c**) Keap1-dependent hyperkeratosis in the esophageal epithelium of *Keap1* genetically modified mice.

**Figure 6 cancers-14-04702-f006:**
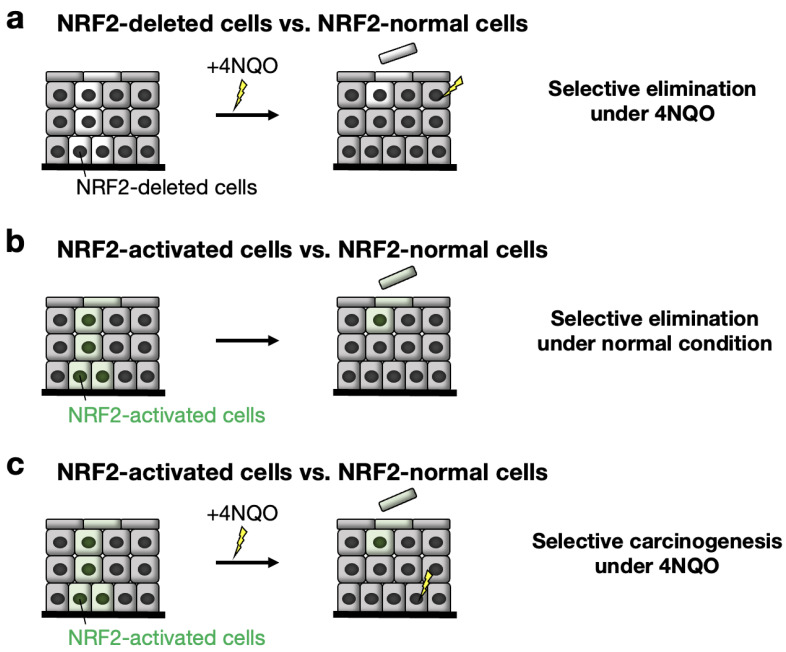
Fate of NRF2-deleted and NRF2-activated cells in the esophageal epithelium when colocalized with NRF2 and KEAP1 normal epithelial cells. The fate of these esophageal cells was examined after treatment with the carcinogen 4NQO (4-nitroquinoline-1-oxide). The results are well explained by the cell competition model (see text). (**a**) NRF2-deleted cells vs. NRF2-normal cells under 4NQO. Note that NRF2-deleted cells are eliminated from the esophageal epithelium under 4NQO administration. (**b**) NRF2-activated cells vs. NRF2-normal cells under 4NQO untreated conditions. Note that NRF2-activated cells are eliminated from the esophageal epithelium. (**c**) NRF2-activated cells vs. NRF2-normal cells under 4NQO. Note that NRF2-activated cells are eliminated from the esophageal epithelium. NRF2-normal cells are an origin of carcinogenesis under 4NQO administration.

**Figure 7 cancers-14-04702-f007:**
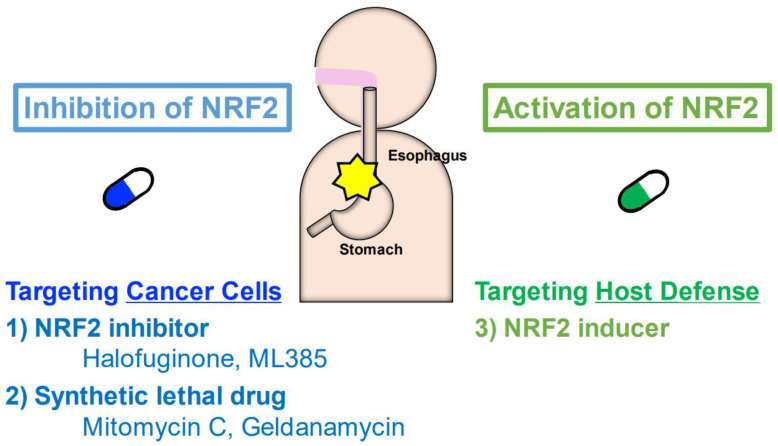
Three new strategies for the treatment of NRF2-addicted cancer. First, NRF2 inhibitors, including halofuginone and ML385, can act to target NRF2-addicted cancer cells. Second, synthetic lethal drugs will target NRF2-addicted cancer cells; this group includes mitomycin C and geldanamycin. Third, NRF2 inducers can activate the host defense system surrounding NRF2-addicted cancers.

**Table 1 cancers-14-04702-t001:** Classification of ESCC cell lines in NRF2 activation. n.d., not determined.

Cell Line	Differentiation Stage of Original Cancers	*NRF2* Mutation	*KEAP1* Mutation	Other Mutation	References
NRF2-high level					
KYSE70	Poor	W24C (homo)	None		[21,79,80,81,82,83]
KYSE110	Poor	E82D (hetero)	n.d.		[21]
KYSE180	High	D77V (homo)	P278Q (hetero)		[21,82,83,84]
KYSE510	High	None	None	PIK3CA E545K (hetero)	[78,82,83]
KYSE520	Moderate	T80I (homo)	None		[82,83]
OE21	Moderate	G81S (hetero) D318H (hetero)	None		[82,83]
TE6	High	F71_D77del (homo)	None		[73,82,83]
TE9	Poor	None	None		[73,82,83]
TE11	Moderate	D29G (homo)	G9R (homo)		[73,82,83]
TE14	Moderate	n.d.	n.d.		[73]
NRF2-normal level					
COLO680N	n.d.	None	None		[82,83]
ECGI10	n.d.	n.d.	n.d.		[82]
KYSE30	High	n.d.	n.d.		[73,78,81]
KYSE140	Moderate	None	None		[82,83]
KYSE150	Poor	None	None		[78,82,83,84]
KYSE270	High	None	None		[82,83]
KYSE410	Poor	None	None		[78,79,82,83]
KYSE450	High	None	c.1708+2_1709del (hetero)		[82,83]
TE1	High	None	None		[73,78,82,83]
TE4	High	None	None		[73,82,83]
TE5	Poor	None	None		[82,83]
TE8	Moderate	None	None		[73,82,83]
TE10	High	None	None		[73,82,83]
TE15	High	None	None		[73,82,83]

**Table 2 cancers-14-04702-t002:** Esophageal phenotypes of genetically modified mouse models targeting *Nrf2* and *Keap1*.

Mouse	Reference			Esophageal Phenotype
*Nrf2* ^–/–^	Chan et al.	1996	[85]	No obvious phenotypes
	Itoh et al.	1997	[86]	
*Nrf2*^flox/flox^::*K5CreERT2*	Horiuchi et al.	2021	[88]	
*Keap1* ^–/–^	Wakabayashi et al.	2003	[52]	Severe hyperkeratosis (juvenile lethality)
*Keap1*^–/–^::*Nrf2*^–/–^				Rescued hyperkeratosis
*Keap1*^–/–^::*Nrf2*^+/–^	Suzuki et al.	2013	[89]	Mild hyperkeratosis (survival) Smaller body than *Keap1*^–/–^::*Nrf2*^–/–^ mice
*Keap1*^–/–^::*Nrf2*^flox/flox^::*K5Cre* (NEKO)	Suzuki et al.	2017	[90]	Rescued but kidney defect
*Keap1*^–/–^::*Tg-Keap1*^WT^	Yamamoto et al. Suzuki et al.	2008 2011	[91] [92]	Rescued hyperkeratosis
*Keap1*^–/–^::*Tg-Keap1*^mutant^	Yamamoto et al. Suzuki et al.	2008 2011	[91] [92]	Severe hyperkeratosis (juvenile lethality)
*Keap1*^–/–^::*Tg-Keap1*^WT^::*Tg-Keap1*^mutant^	Suzuki et al.	2011	[92]	Severe hyperkeratosis (juvenile lethality)
*Keap1* ^floxA/+^	Taguchi et al.	2010	[93]	No obvious phenotype
*Keap1* ^floxA/–^				Mild hyperkeratosis (survival)
*Keap1*^floxA/–^::*Tg-Keap1*^WT^				Rescued hyperkeratosis
*Keap1*^floxA/+^::*K5Cre*				No obvious phenotype
*Keap1*^floxA/floxA^::*K5Cre*				Severe hyperkeratosis (juvenile lethality)
*Keap1* ^floxB/floxB^	Blake et al.	2010	[94]	No obvious phenotype
*Keap1*^floxB/floxB^::*K5CreERT2*	Hirose et al.	2022	[95]	Atypical phenotype

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
