# Peer review of "The KEAP1-NRF2 System and Esophageal Cancer"

_cancers, 2022, doi:10.3390/cancers14194702_

Round 1
Reviewer 1 Report
In this manuscript by Hirose et al., the authors summarized the relative data and knowledge of the KEAP1-NRF2 system and the treatment of ESCC. They also discussed the potential strategies for the treatment of NRF2-addicted cancer. The latest research progress was reviewed. This review provided interesting information and may help future scientific research and clinical application.
Author Response
We would like to thank the reviewer for the efforts and understanding.
Reviewer 2 Report
This is a very interesting and well written review. To my opinion only few points deserve to be implemented. In particular:
- Regulatory Mechanisms of NRF2 Activation: authors should underline that several molecules are known to be a Nrf2 inducers, such as sulforaphane, dimethyl fumarate and bardoxolone methyl (PMID: 33123312).
- Line 122: authors correctly state that NRF2-addicted cancers acquire a marked advantage in proliferation and resilience against chemoradiotherapy, but they should also specify that a strategy targeting Nrf2 activation has been proposed for cancer treatment (PMID: 35453297).
- 9.1. NRF2 Inhibitors to Treat NRF2-Addicted Cancers; It deserves to be specified that many compounds have shown an inhibitory effect on NRF/KEAP1 pathway in different types of cancers (PMID: 35901941, 31032633, 31396308). This is an important point to add since the compounds analysed by the authors may not be the only NRF2 inhibitors in ESCC then stimulating further research in this research field.
- Figure 6a: correct "NRF2-deleded" with "NRF2-deleted" and "nomal" with "normal"
- The abbreviations in figures should be reported in full length in the figure legend
Author Response
This is a very interesting and well written review. To my opinion only few points deserve to be implemented. In particular:
We thank the reviewer for the constructive comments. We have revised our manuscript following the advice.
- Regulatory Mechanisms of NRF2 Activation: authors should underline that several molecules are known to be a Nrf2 inducers, such as sulforaphane, dimethyl fumarate and bardoxolone methyl (PMID: 33123312).
Following the reviewer suggestion, we have described Nrf2 inducers in the main text. We surmise it is much informative describing Nrf2 inducers in Section 9.3 “A NRF2 Inducers to Target the Host Defense System”, rather than in Section 1, so we have added the descriptions to Lines 516–520.
- Line 122: authors correctly state that NRF2-addicted cancers acquire a marked advantage in proliferation and resilience against chemoradiotherapy, but they should also specify that a strategy targeting Nrf2 activation has been proposed for cancer treatment (PMID: 35453297).
This is an important comment, and we agree. We have added a few sentences describing NRF2 activation for cancer treatment to Line 122–123.
- 9.1. NRF2 Inhibitors to Treat NRF2-Addicted Cancers; It deserves to be specified that many compounds have shown an inhibitory effect on NRF/KEAP1 pathway in different types of cancers (PMID: 35901941, 31032633, 31396308). This is an important point to add since the compounds analysed by the authors may not be the only NRF2 inhibitors in ESCC then stimulating further research in this research field.
We agree with this comment. There are other compounds, which show an inhibitory effect on NRF2. We have added some sentences to Line 468–470.
- Figure 6a: correct "NRF2-deleded" with "NRF2-deleted" and "nomal" with "normal"
We apologize for our oversights and have revised the Figure 6a.
- The abbreviations in figures should be reported in full length in the figure legend.
We have added full length of the abbreviations in Figures.
Reviewer 3 Report
In the present manuscript titled “The KEAP1-NRF2 System and Esophageal Cancer”, authors summarized distinct approaches for the treatment of NRF2 mediated ESCC. Authors described the KEAP1-NRF2 system very well and their involvement with esophageal squamous cell carcinoma (ESCC) and esophageal adenocarcinoma (EAC). However, there are some points need to be incorporated.
11. Abstract is well written and conclusive.
22. Describe the role of miRNA involved in NRF2-KEAP1 mediated regulation of ESCC (for example: DOI: 10.1158/1541-7786.MCR-17-0232 ; DOI: 10.1158/1541-7786.MCR-13-0246-T).
33. Conclusion should be elaborated with future direction.
44. References are not in a constant format.
Author Response
In the present manuscript titled “The KEAP1-NRF2 System and Esophageal Cancer”, authors summarized distinct approaches for the treatment of NRF2 mediated ESCC. Authors described the KEAP1-NRF2 system very well and their involvement with esophageal squamous cell carcinoma (ESCC) and esophageal adenocarcinoma (EAC). However, there are some points need to be incorporated.
We thank the reviewer for the important comments.
11. Abstract is well written and conclusive.
We appreciate for this professional comment.
22. Describe the role of miRNA involved in NRF2-KEAP1 mediated regulation of ESCC (for example: DOI: 10.1158/1541-7786.MCR-17-0232 ; DOI: 10.1158/1541-7786.MCR-13-0246-T).
We thank the reviewer for this important comment. miRNA is an interesting approach to target NRF2-addicted ESCC. We have added a few sentences to Line 531–535.
33. Conclusion should be elaborated with future direction.
We thank for this important comment. We have revised the conclusion and included the future direction.
44. References are not in a constant format.
We have revised the format.